# Oxygen Vacancies on Hydrated Anatase (101) Surfaces: Insights from Classical and Ab Initio Molecular Dynamics Simulations

**DOI:** 10.3390/nano15050364

**Published:** 2025-02-27

**Authors:** Fredrik Grote, Alexander Lyubartsev

**Affiliations:** Deparment of Chemistry, Stockholm University, Svante Arrhenius väg 16 C, 10691 Stockholm, Sweden; fredrik.grote@su.se

**Keywords:** titanium dioxide nanomaterials, oxygen vacancies, molecular dynamics simulations

## Abstract

Hydrated anatase (101) titanium dioxide surfaces with oxygen vacancies have been studied using a combination of classical and ab initio molecular dynamics simulations. The reactivity of surface oxygen vacancies was investigated using ab initio calculations, showing that water molecules quickly adsorb to oxygen vacancy sites upon hydration. The oxygen vacancy then quickly reacts with the adsorbed water, forming a protonated bridging oxygen atom at the vacancy site and at a neighboring oxygen bridge. Ab initio simulations also revealed that this occurs via a short-lived hydronium ion intermediate. It was investigated how this reaction affects the structure and dynamics of water near the anatase surface. Classical molecular dynamics simulations of surfaces with and without oxygen vacancies showed that vacancies disrupt the second solvation shell, consisting of water molecules hydrogen bonded to the surface, thereby changing the local water density and diffusion as well as the binding modes for hydrogen bonding. Our findings support the hydroxylation of oxygen vacancies on anatase (101) surfaces, rather than stabilization by molecular adsorption or subsurface diffusion. The work gives new atomistic insight into water structure and surface chemistry on the catalytically relevant anatase (101) titanium dioxide surface.

## 1. Introduction

Titanium dioxide (${\mathrm{TiO}}_{2}$) nanomaterials stand out as one of the most useful systems in nanotechnology [1]. This metal oxide occurs in several polymorphic forms [2,3], and the most common ones are rutile, anatase, and brookite [4], as bulk material rutile is the thermodynamically stable form, while anatase is more stable for particles with sizes smaller than about 14 nm [5]. Applications of ${\mathrm{TiO}}_{2}$ include solar cells [6], white pigment [7] in paints and cosmetics, UV-filter in sunscreen lotions [8], and self-cleaning surface coatings [9]. ${\mathrm{TiO}}_{2}$ has also received a significant amount of interest in the context of photocatalytic water splitting as a way of producing hydrogen from water using clean energy from sunlight [10]. However, the wide band gap of ${\mathrm{TiO}}_{2}$ (>3 eV) makes it adsorb in the UV range, and to make water splitting more efficient, it needs to be shifted to the more intense VIS part of the solar spectrum, e.g., by doping or introducing defects [11]. Applications of ${\mathrm{TiO}}_{2}$ have been subject to several extensive reviews [12,13]. With these challenges in mind, it is of interest to understand how the surface structure of titanium dioxide relates to its reactivity and surface chemistry. In particular, anatase is the titanium dioxide polymorph that shows the highest catalytic activity [14]. The anatase (101) surface has the lowest surface energy and is also the one dominating in nanomaterial samples [15]. It exposes oxygen atoms that are 2-coordinated to titanium and titanium atoms that are 5-coordinated to oxygen. This can be compared with the situation in bulk, where oxygen atoms are 3-coordinated and titanium atoms are 6-coordinated. Surface structure and modifications affect the electronic structure of the material and may alter its catalytic efficiency. For example, reactions often occur at corners or edges where atoms are exposed and exist in lower coordination environments than in bulk or at an ideal plane surface [16]. Furthermore, the presence of defects including oxygen vacancies, where 2-coordinated oxygen sites ($O_{2c}$) are vacant, has been shown to affect the reactivity of these materials [17,18]. Some titanium dioxide surfaces have been reported to have oxygen vacancies on as much as 15% of all $O_{2c}$ sites [19]. Oxygen deficiency leads to the formation of titanium suboxides, e.g., Magneli phases, and these materials have received significant interest due to their electrochemical and photocatalytic properties [20].

Oxygen vacancies on the anatase (101) surface have been investigated in several experimental studies. Ref. [21] studied several titanium dioxide surfaces using resonant photoemission and X-ray adsorption spectroscopy and reported that anatase (101) has a higher concentration of oxygen vacancies than both anatase (001) and rutile (110). This was proposed as an explanation for the higher catalytic activity of anatase compared to rutile. Scanning tunneling microscopy (STM) experiments on the other hand, providing a more direct detection of oxygen vacancies, indicate low amount of vacancies on newly cleaved anatase (101) surfaces [22]. However, oxygen vacancies can be introduced on titanium dioxide surfaces by electron bombardment in order to alter their catalytic activity [23]. Of note, an extensive study was reported in ref. [24], where oxygen vacancies were first introduced on the anatase (101) surface by electron bombardment at 105 K and then probed using STM after equilibration at different temperatures. It was found that the fraction of vacancies remaining on the surface after equilibration starts to decrease for temperatures > 200 K and reaches zero at about 450 K (at 300 K roughly 60% of vacancies remained). This result is attributed to a temperature-dependent equilibrium, where oxygen vacancies diffuse between the surface and the subsurface, where they are more stable. The higher stability of vacancies in the bulk compared to the surface can be understood, considering that every vacancy on the surface involves the formation of one 5-coordinated and one 4-coordinated Ti atom, while the formation of a vacancy in the bulk leads to two 5-coordinated Ti atoms. These findings are supported by the theoretical results of Cheng and Selloni [25], who carried out a detailed study of vacancy formation energies at surface and subsurface sites using DFT calculations. Their results showed that subsurface vacancies are about 0.5 eV more stable compared to those on surface sites. This indicates that a vast majority of oxygen vacancies exist in the subsurface/bulk region, which could explain the high vacancy concentration obtained from resonant photoemission and X-ray adsorption spectroscopy (i.e., ref. [21]), which do not distinguish between the surface and subsurface vacancies. Cheng and Selloni also calculated the surface to subsurface diffusion barrier to be about 0.7 eV.

It should be noted that STM studies [22,24] of titanium dioxide surfaces are carried out under vacuum conditions. Also, DFT calculations [25] studying the energetics of vacancy formation and their diffusion barriers between surface and subsurface regions were performed without a solvent. These results may therefore differ from the situation for surfaces exposed to water, which are of interest for many of the mentioned applications. Fisicaro et al. [26] carried out DFT calculations of formation energies for surface and subsurface oxygen vacancies in an anatase (101) slab in vacuum and in contact with solvents (water and ethanol). Consistent with ref. [25], they reported that subsurface vacancies are more stable than those on the surface for the slab in vacuum (with energy difference of 0.4 eV). Interestingly, they found that the situation is the opposite when the surface is hydrated. For the hydrated slab surface, vacancies are 0.62 eV more stable than those in the subsurface region. They also carried out ab initio molecular dynamics (AIMD) simulations and observed that vacancies diffuse from the subsurface to the surface, indicating a low-diffusion barrier. Water surface reactions are known to modify titanium dioxide–water interfaces [27]. Water molecules can adsorb molecularly to surface Ti atoms (i.e., as intact water molecules) or dissociatively (i.e., forming hydroxyl groups and protonated oxygen bridges). It seems plausible that water molecules would react with oxygen vacancies on the surface forming hydroxyl groups or protonated oxygen bridges, thereby eliminating unfavorable 4-coordinated titanium atoms. Alternatively, water molecules may adsorb molecularly and thereby stabilize the vacancies on the hydrated surface. Fisicaro et al. [26] reported that while ethanol dissociates, adsorbed water molecules coexist with the surface oxygen vacancy due to the small energy difference between molecular and dissociative adsorption. However, the production part of their AIMD simulation was short (5 ps), and sampling these reactive events often requires simulations on over ∼10 ps time scale [16]. This raises the question about the fate of oxygen vacancies on hydrated anatase (101) surfaces: are they stabilized by molecularly adsorbed water or do they become hydroxylated due to dissociative water adsorption?

Obtaining a detailed picture of the water structure and surface chemistry at titanium dioxide–water interfaces in general, and the anatase (101) surface in particular, is a central challenge in several scientific fields, including the modeling, surface characterization, and synthesis communities. It is also crucial for better understanding the catalytic processes taking place in these systems, particularly photocatalytic water splitting and other applications. It is also important for understanding the effect of these materials on biological systems, as well as on environment where they exist in aqueous solution, to ensure their safety. Here, simulations can provide important atomistic information that can complement and help the interpretation of experimental studies. The aim of the present work is two-fold: (1) investigate the fate of oxygen vacancies, i.e., subsurface diffusion, water adsorption or hydroxylation using AIMD, and (2) explore, for the first time, how this affects the structure and dynamics of water at the solid–liquid interface. We therefore performed a 30 ps AIMD simulation of a hydrated anatase (101) surface with an oxygen vacancy in order to observe possible reactions between the vacancy and water. Studying the structure and dynamics of water at the interface requires larger systems and longer simulation times than what is accessible for AIMD. For this reason, we also performed classical MD (CMD) simulations of the anatase (101) surfaces with and without oxygen vacancies on the surface. In the next section ‘Materials and Methods’, we describe the simulations that were performed. In the ‘Results’ section, we describe and discuss the simulation results, and in the ‘Discussion’, we provide our interpretations. The final section ‘Conclusions’ summarizes our findings and gives our concluding remarks.

## 2. Materials and Methods

### 2.1. AIMD Simulations


In order to study water reactivity at an oxygen vacancy on the hydrated anatase (101) surface, we performed an AIMD simulation of a slab with thickness 9 Å and widths 10 × 11 Å. The total z-length of the simulation box was 40 Å, giving a water layer with a thickness of about 15 Å; see Figure 1a. One $O_{2c}$ atom was removed, forming an oxygen vacancy on the surface; see Figure 1b. The simulation was carried out using the Gaussian and plane wave method [28] (GPW) as implemented in the QUICKSTEP module [29] of the CP2K software [30]. We used the BLYP exchange-correlation functional [31,32] augmented with the D3 dispersion correction developed by Grimme [33]. The MOLOPT-DZVP basis set [34] was used together with GTH pseudopentials [35]. In addition to the Gaussian basis set, the GPW method uses a dual plane wave basis set allowing for the efficient calculation of the Hartree energy. We used a 300 Ry plane wave cutoff, and the relative cutoff was 30 Ry. This choice was based on a series of test calculations, where we monitored the convergence of the total energy as well as the distribution of Gaussians on the multigrid; see Appendix A. The system was simulated in the NVT ensemble, keeping the temperature constant using the thermostat by Bussi, Donadio, and Parrinello [36] with a time constant of 0.1 ps. Periodic boundary conditions were applied in the x-, y-, and z-dimension, and the time step was 0.5 fs. In total, the system consisted of 455 atoms and was simulated for 30 ps.

### 2.2. CMD Simulations

The structure of water near an anatase (101) surface with oxygen vacancies was studied by classical MD simulations. We performed a simulation of a slab having a width of about 31 Å × 30 Å and thickness 31 Å. For titanium dioxide, we used a force field [37] that had previously been developed in our group, and water was described using the TIP3P model. Hydroxyl groups were added to the surface Ti atoms in order to give a surface charge −$0.56\;{\mathrm{e/nm}}^{2}$ corresponding to the measured zeta-potential at neutral pH [38]. In order to neutralize the system, $11\;{\mathrm{Na}}^{+}$ ions were added. These ions are present both in biological systems and in buffer solutions, keeping neutral pH. Oxygen vacancies were introduced on the surface by removing $14\;O_{2c}$ atoms (corresponding to 15% of the surface oxygens). This gave the system a net charge of about +14.5, which was removed from bulk Ti atoms in order to keep the charge of the slab the same. This had a minimal effect, changing the net charge of atom type TiA from +2.24 to +2.23. The system was hydrated with 3012 water molecules, giving a water layer of about 100 Å; see Figure 1c. Periodic boundary conditions were applied in the x-, y-, and z-dimension. In order to obtain a reasonable starting configuration, we first performed energy minimization using the steepest decent algorithm. We then carried out a 1 ns equilibration, keeping the temperature constant at 300 K using the v-rescale thermostat [36] with a time constant of 1 ps. In order to obtain a realistic density of the system, we then performed a 1 ns equilibration, holding pressure constant at 1 bar using the Berendsen barostat [39] with a time constant of 5 ps. We then performed a 100 ns production simulation in the NVT ensemble, where the first 50 ns was discarded as equilibration. Trajectory frames were saved every 10 ps, but since the analysis of water dynamics requires a higher temporal resolution, we extended the simulations for 100 ps, saving the trajectory every MD step. Electrostatic interactions were computed using the PME algorithm [40] with a 1.2 nm real space cutoff and 0.12 grid spacing. A 1.2 nm cutoff was used for non-bonded interactions with dispersion correction applied to energy and pressure. We used constrained hydrogen bonds and a time step of 2 fs. For comparison, we also carried out a simulation using the same simulations settings for the exact same system but without oxygen vacancies on the surface. All classical MD simulations were performed using the Gromacs software [41].

## 3. Results

### 3.1. Water Reactivity at Oxygen Vacancy

In order to study if water molecules dissociate or become molecularly adsorbed to oxygen vacancies on the anatase (101) surface, we performed an AIMD simulation. We observed the splitting of a water molecule at the site of the oxygen vacancy, forming two protonated oxygen bridges on the surface, one at the previously vacant oxygen site and one at the neighboring oxygen bridge. We now give a detailed description of the mechanism of how this happens. Figure 2 shows the simulated hydrated titanium dioxide slab where the reactive region and the atoms taking part in the reaction are highlighted. It also shows the interatomic distances that are used as reaction coordinates. After introducing the oxygen vacancy, a water molecule quickly adsorbs to one of the undercoordinated Ti atoms at the oxygen vacancy (atoms $O_{153}{\mathrm{-H}}_{151}$ in Figure 2). A second water molecule ($O_{162}{\mathrm{-H}}_{154}$) acts as a hydrogen bond donor, forming a hydrogen bond to the adsorbed water as well as to a neighboring bridging oxygen atom; see configuration (1) in Figure 3. After about 16–17 ps of AIMD simulation, these two waters undergo simultaneous rotations about their O-H bonds. This brings them to structure (2) where the adsorbed water which previously accepted a hydrogen bond now instead donates a hydrogen bond to the second water molecule. These simultaneous rotations are responsible for the fast drop in the $O_{162}{\mathrm{-H}}_{154}$ distance from about 3 to 1.5 Å; see Figure 3. In this configuration, the system undergoes fluctuations, where the adsorbed water molecule forms a protonated oxygen bridge and the second water forms a short-lived hydrogen bonded hydronium ion ($H_{3}O^{+}$) species; see structure (3). This intermediate forms two times but decompose back to configuration (2). The third time the intermediate is formed, it donates one of its protons to the neighboring oxygen bridge, resulting in the splitting of one water molecule and formation of two neighboring protonated oxygen bridges. This structure remains stable during the rest of the simulation.

### 3.2. Water Structure Near Titanium Dioxide Surface with Oxygen Vacancies

Oxygen vacancies can be introduced on anatase (101) surfaces by electron bombardment. However, our AIMD simulations showed that the subsequent hydration of the surface leads to the dissociation of water molecules at the vacancy sites, forming protonated oxygen bridges. How does this reaction, i.e., the removal of vacancies from the surface, change the structure of water at the interface? From our CMD simulations with and without oxygen vacancies, we calculated water number density profiles along the z-direction (i.e., direction normal to the surface), informing about the water structure at the solid–liquid interface. Figure 4 shows water density as function of the z-distance for the ideal surface and the surface with oxygen vacancies present. For the ideal surface, the density profile shows two strong maxima: one, just above 2 Å, corresponding to molecularly bound water (see Figure 4b), and another between 3 and 4 Å from waters forming hydrogen bonds to bridging oxygen atoms on the surface (see Figure 4c). When oxygen vacancies are present, the situation is different, and two additional peaks appear in the density profile. The first one located at distance 1–2 Å corresponds to water adsorbed at oxygen vacancy sites (shown in Figure 4d). The second one appears just below 3 Å in between the peak corresponding to molecularly bound water and hydrogen bonded water. Interestingly, this peak is actually also present in the density profile for the ideal surface (but as a very weak feature). This indicates the presence of two binding modes for hydrogen bonded water. It is suggested that the one observed at a shorter distance (z<3 Å) corresponds to water molecules donating two hydrogen bonds to bridging oxygen atoms on the titanium dioxide surface, whereas the peak at longer distance (z>3 Å) is due to waters forming only one hydrogen bond to the surface. On the ideal surface, the longer single hydrogen bond mode dominates, while the presence of oxygen vacancies on the surface enhances the binding mode with two hydrogen bonds. This shows that the structure of water near the surface is sensitive to the presence of defects on the surface. It is therefore likely that it also affects the dynamics of water molecules near the interface.

### 3.3. Water Dynamics Near Titanium Dioxide Surface with Oxygen Vacancies

In order to gain insight into how the dynamics of water near the anatase (101) surface is affected by oxygen vacancies, we calculated the mean square displacement (MSD) for oxygen atoms in water molecules in shells located at different z-distance from the surface. The mean square displacement is related to the lateral diffusion coefficient according to Equation (Equation 1):(1)$$D=\lim_{\tau \to \infty }\frac{1}{4}\frac{d}{d\tau }\langle |\vec{r}\left(t+\tau \right)-\vec{r}\left(t\right){|}^{2}\rangle$$
where $\vec{r}\left(t\right)$ is the position vector of a water oxygen atom at an initial time *t*, and $\vec{r}\left(t+\tau \right)$ is its position at a later time t+τ. In order to obtain information on how diffusion depends on the distance from the titanium dioxide surface, we calculated MSD for water molecules in regions between distance *z* and z+dz, where dz=0.5 Å, for water molecules present in the region both at time *t* and time t+τ. The computed MSD as a function of lag time are shown in Figure 5. From the MSD curves for the ideal surface (see Figure 5a), it is seen that water diffusion slows down with decreasing distance to the surface. The same trend can be noticed for the surface with oxygen vacancies (see Figure 5b). However, for z>3 Å, diffusion is faster for the surface with oxygen vacancies compared to the ideal surface. This is most likely explained by hydrogen bonds being disrupted by the presence of oxygen vacancies. This lowers the density of water at this distance (see density profile in Figure 4a) and makes diffusion faster.

The MSD informs about the lateral dynamics of water molecules in regions located at different distances along the *z*-axis. It is also interesting to analyze the dynamics of water molecules between the different layers. This can be performed by calculating the survival probability (SP) of waters in these layers according to Equation (Equation 2):(2)$$P\left(\tau \right)=\langle \frac{N(t+\tau )}{N(t)}\rangle$$
where N(t) is the number of water molecules in a region at an initial time *t*, and N(t+τ) is the number of those water molecules that remain in the region at a later time t+τ. Figure 6 presents the survival probabilities from water molecules in layers between *z* and z+dz. The result for the ideal surface (see Figure 6a) shows that water molecules close to the surface (z<4 Å) have high SP (>0.6 up to 30 ps), while the SP of molecules further out from the surface is lower (decays down to 0.1–0.3). This illustrates that water molecules closer to the surface, due to their interactions with the titanium dioxide surface, are less mobile compared to those further out. The same trend with SP decreasing with the distance to the surface can be seen for the surface with oxygen vacancies (see Figure 6b). While the SP at larger distances (z>3.5 Å) is essentially identical to the result obtained for the ideal surface, differences can be noticed at a shorter distance. For z<3.5 Å, the SP decays faster for the surface with defects compared to the ideal surface. This shows that also the dynamics between layers (i.e., along the z-direction) is faster when defects are present on the surface. The oscillations in the SP curves for the ideal surface at a short distance (z<3.5 Å) show that the same water molecules leave and then re-enter the region (after about 10 ps). On the defect surface, water molecules are more mobile, and the SP curves decay also for short distances from the surface.

## 4. Discussion

Introducing oxygen vacancies on anatase (101) surfaces, e.g., by electron bombardment, affects the reactivity as well as the structure and dynamics of water near the solid–liquid interface where catalytic reactions take place. While oxygen vacancies are more stable in bulk compared to the surface region under vacuum conditions, due to the formation of 4-coordinated Ti atoms on the surface, hydration changes the picture. On the hydrated surface, it is energetically favorable for oxygen vacancies to exist on the surface due to interaction with the surface water. However, for vacancies to be stable on the hydrated surface, it is required that water molecules adsorb molecularly, thereby stabilizing the unfavorable 4-coordinated Ti atoms. If instead water molecules dissociate, this will replace the vacancies and hydroxylate the surface. Our AIMD simulation favors the latter scenario, showing that water quickly adsorbs to the vacancy site and then dissociates, leading to the formation of two protonated oxygen bridges, one at the initially vacant site and one at a neighboring oxygen bridge. The reaction takes place via a short-lived hydronium ion intermediate. This happens already after 17 ps of simulation, which suggests that oxygen vacancies on hydrated anatase (101) are not stable. Another possibility is that there is an equilibrium between hydroxylated vacancies and intact vacancies stabilized by adsorbed water. However, we do not observe the reverse process (i.e., vacancy reforming after hydroxylation). Here, it should be noted that our AIMD simulation is only 30 ps, which could be too short to sample these processes. Our simulation results support a mixed picture, where water molecules both adsorb molecularly and dissociatively to the anatase (101) surface, which is in agreement with surface-sensitive photoelectron spectroscopy experiments [42]. We point out further that our results are dependent on the quantum chemical theory that have been used. In particular, it is known that GGA functionals give the too delocalized description of electrons in ${\mathrm{TiO}}_{2}$ as well as the too strong hydrogen bonding in water [43,44]. We use an adiabatic picture, where the system remains on the ground state energy surface during reactions. While this is usually a good approximation for situations where the system is close to its equilibrium geometry, it should be pointed out that it can break down during reactions when states couple and their potential energy surfaces come close to each other. Furthermore, nuclear quantum effects are assumed to be unimportant, which can be questionable especially for protons having small mass. These effects are captured by path integral simulations.

Interactions between water and ${\mathrm{TiO}}_{2}$ strongly impact the structure and dynamics of water near the surface. Our classical MD simulations of the ideal anatase (101) surface and the same surface with 15% oxygen vacancies show that the presence of vacancies strongly affects the structure of water near the surface. In particular, oxygen vacancies disrupt the hydrogen bond network and favor a binding mode, where water donates two hydrogen bonds to ${\mathrm{TiO}}_{2}$. We observe that the local density of water at distance about 3 Å from the surface (i.e., in the hydrogen bond layer) decreases. Changes in the hydrogen bond structure have further consequences on the dynamics of water near the anatase surface. The MSD analysis of water molecules in shells located at different distances along the surface normal show that the lateral dynamics of water is faster with oxygen vacancies present. This is a direct consequence of the disruption of the hydrogen bond layer decreasing the local water density. SP analysis shows that also the dynamics of water along the surface normal becomes faster due to the presence of oxygen vacancies. The lower local water density and faster dynamics near the solid–liquid interface make the surface more accessible for adsorbate/reactant molecules, which can be of importance for applications in catalysis. However, it should be pointed out that our results can be sensitive to our choice of force field describing interactions in the simulated system. We use a fixed-point charge representation of the molecular charge distribution, which does not account for polarizability. Such effects can be important particularly when modeling interfaces where the charge distribution needs flexibility to adjust as molecules move between bulk and surface regions. This can be investigated using polarizable force fields.

## 5. Conclusions

In this work, we have studied oxygen vacancies on the hydrated catalytically relevant anatase (101) surface using a combination of ab initio and classical molecular dynamics simulations. We found that the vacancy quickly reacts with water forming protonated oxygen bridges, indicating that oxygen vacancies are not stable on the surface. Our results also show that the presence of defects strongly influences the structure of the solid–liquid interface. We found that the local water density is significantly lowered in the second solvation shell (corresponding to hydrogen bonded water) when oxygen vacancies are present on the surface. This is due to the hydrogen bond network getting disrupted, splitting into two density peaks that are attributed to two distinct binding modes where water donates one and two hydrogen bonds to the titanium dioxide surface. The disruption of the hydrogen bond network in the second solvation shell has implications on the dynamics of water near the surface. Oxygen vacancies makes both the lateral dynamics of water and the dynamics along the surface normal faster. Future works can investigate how the result depends on the level of quantum chemical theory, particularly the choice of the exchange-correlation functional. Results from classical MD simulations are highly dependent on the force field used, and in order to gain further confidence, future works can investigate the effect of using different models. Our results highlight the importance of structural defects for the properties of metal oxide surfaces. This work has focused on a particular type of defect, oxygen vacancies, but studies of other types of defects including interstitials and surface steps can give a more complete picture and potentially play an important role in guiding defect engineering aiming to enhance the catalytic performance of ${\mathrm{TiO}}_{2}$ nanomaterials.

## Figures and Tables

**Figure 1 nanomaterials-15-00364-f001:**
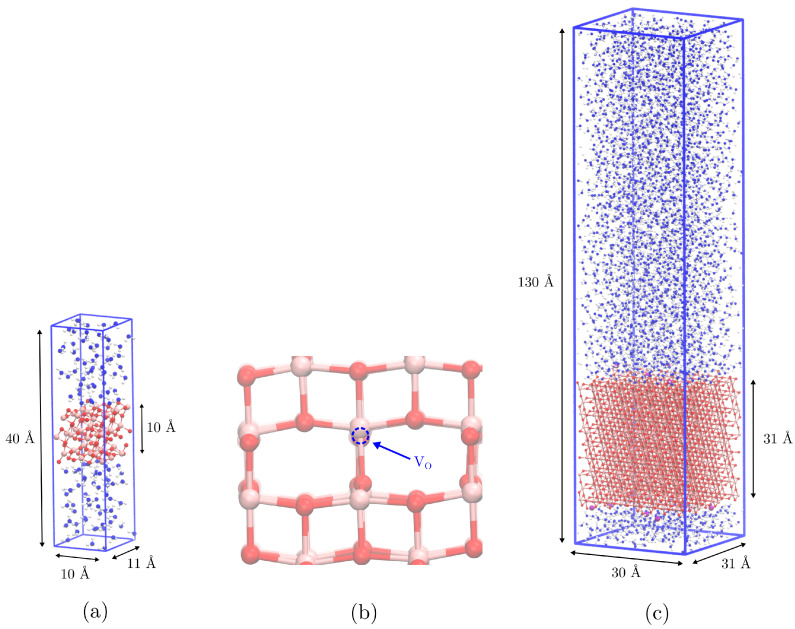
Snapshots showing (**a**) system from AIMD simulation, (**b**) oxygen vacancy on the surface, and (**c**) system from CMD.

**Figure 2 nanomaterials-15-00364-f002:**
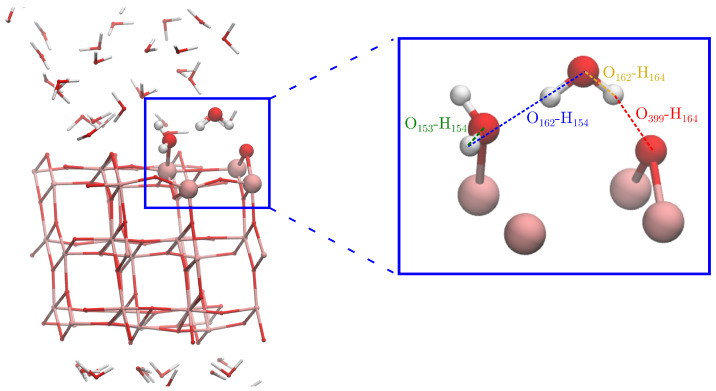
Hydrated titanium dioxide slab from the AIMD simulation with the reactive region containing the oxygen vacancy highlighted.

**Figure 3 nanomaterials-15-00364-f003:**
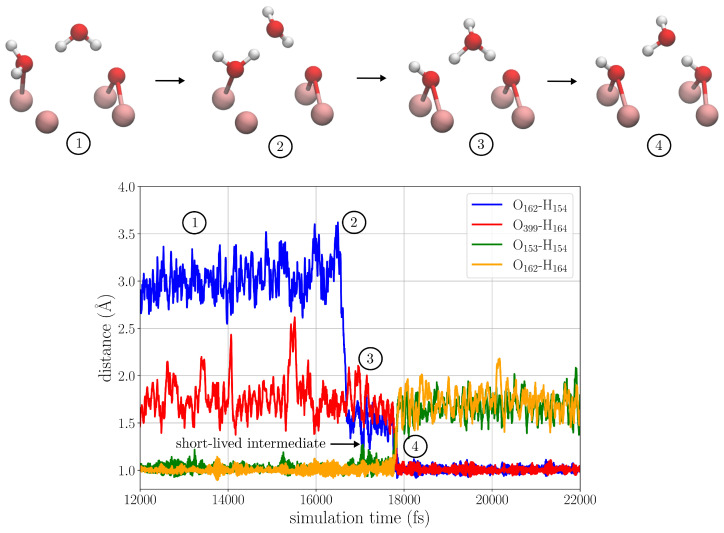
Reaction coordinates as function of time during water splitting at oxygen vacancy.

**Figure 4 nanomaterials-15-00364-f004:**
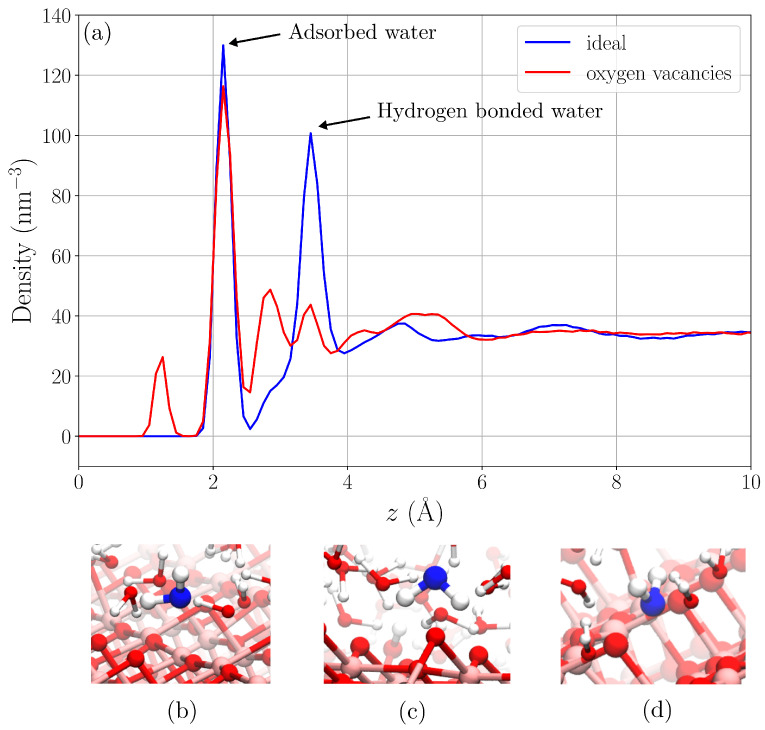
Water density profile (**a**) and water binding modes from MD simulations (**b**–**d**).

**Figure 5 nanomaterials-15-00364-f005:**
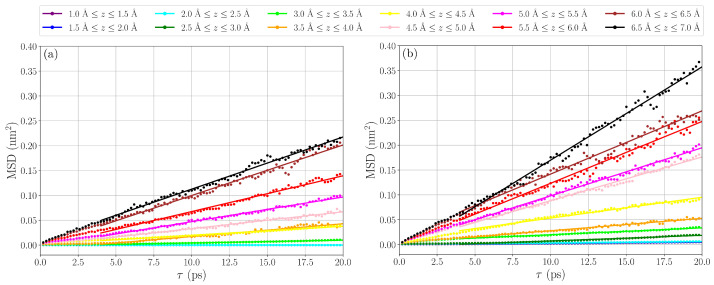
MSD of oxygen atoms in water located at different z-distance from the surface for (**a**) ideal surface and (**b**) surface with oxygen vacancies.

**Figure 6 nanomaterials-15-00364-f006:**
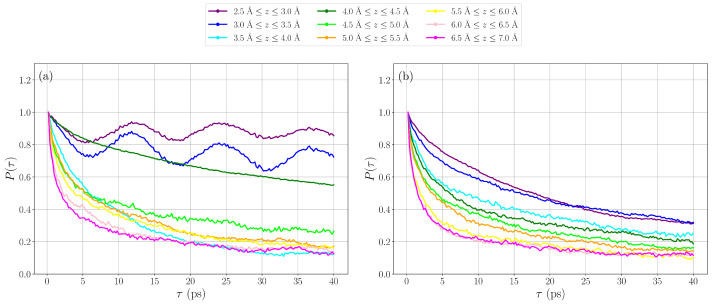
Survival probability for water oxygen atoms located at different z-distances from the surface for (**a**) the ideal surface and (**b**) the defect surface.

## Data Availability

Data are available upon reasonable request.

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
