# Peer review of "Oxygen Vacancies on Hydrated Anatase (101) Surfaces: Insights from Classical and Ab Initio Molecular Dynamics Simulations"

_nanomaterials, 2025, doi:10.3390/nano15050364_

Round 1
Reviewer 1 Report
Comments and Suggestions for Authors
Titanium dioxide plays a significant role in catalytic applications, such as photocatalytic water splitting for hydrogen production, self-cleaning surfaces, and UV antibacterial protection. Understanding how surface defects like oxygen vacancies influence TiO₂'s catalytic performance is actual for optimizing these applications. Therefore, this study is interesting for the readers, especially a computational community.
Nevertheless, several aspects of the study remain unclear to me or require explanation before publication.
1) AIMD simulations were conducted for 30 picoseconds. Is it enough to “catch” reverse reactions (i.e., re-formation of vacancies after hydroxylation) essential in real photocatalytic systems?
2) Please provide in the supplementary materials the basis sets used in the study.
3) If I correctly understand, the CMD uses a fixed point charge model, which does not account for polarizability. If yes, how does this simplification impact the study results?
4) It would be beneficial to develop an electrochemical model based on the computational results to describe the energetics of the process (Possibly not in this study, but at least recommendations for further investigation could be added in the discussion part)
5) In principle, it would be useful to include ideas for experimental verification of the results in the discussion section. The study currently focuses on a comprehensive analysis of the model, but at times it lacks connections to real experimental data, which would strengthen the validity of the findings. Could in situ XAS, for example (or other methods) be used to verify the results?
6) line 135: “In order to neutralize the system 11 Na+ ions were added”. This is a new approach for me. Please justify its validity in the text. To what extent could this approach introduce artifacts? Why can't H⁺ (not Na+) ions be added?
Author Response
Reviewers comment: Titanium dioxide plays a significant role in catalytic applications, such as photocatalytic water splitting for hydrogen production, self-cleaning surfaces, and UV antibacterial protection. Understanding how surface defects like oxygen vacancies influence TiO₂'s catalytic performance is actual for optimizing these applications. Therefore, this study is interesting for the readers, especially a computational community.
Nevertheless, several aspects of the study remain unclear to me or require explanation before publication.
1) AIMD simulations were conducted for 30 picoseconds. Is it enough to “catch” reverse reactions (i.e., re-formation of vacancies after hydroxylation) essential in real photocatalytic systems?
Authors reply: We thank the reviewer for assessing our work as interesting for readers in the computational community. The reviewer has a good point that the length of the simulation is not long enough to catch reverse reactions. However, here we are limited by the high computational cost of ab initio MD simulations making it difficult to simulate longer than ~10 ps. Classical MD allows us to study longer time scales but these simulations use fixed pre-defined chemical bonds and therefore do not inform about reactions. To reach time scales longer than what is possible with DFT-based ab initio MD, while still allowing formation/breaking of chemical bonds, one can consider using some cheaper method, e.g. tight binding DFT or reactive force fields. We hope to explore such approaches in the future. In the revised manuscript we included a reservation (page 9, line 277-280) that the simulation time may be limited to catch reverse reactions and reach equilibrium.
Reviewers comment: 2) Please provide in the supplementary materials the basis sets used in the study.
Authors reply: We thank the reviewer for the valuable suggestion and we have now added information about basis set to the supplementary materials (page S1).
Reviewers comment: 3) If I correctly understand, the CMD uses a fixed point charge model, which does not account for polarizability. If yes, how does this simplification impact the study results?
Authors reply: The reviewer is right that our CMD study did not include polarizability which can be important when modeling TiO2-water interfaces. This is one of the limitations with our CMD simulations. We have therefore pointed out in the manuscript that our CMD simulations did not include polarizability and more clearly stated that it can be investigated using polarizable force fields, see Discussion section (page 10, line 313-314) of the revised manuscript:
“We have used a fixed point charge representation of the molecular charge distribution which does not account for polarizability. Such effects can be important in particular when modelling interfaces where the charge distribution needs flexibility to adjust as molecules move between bulk and surface regions. This can be investigated using polarizable force fields.”
Reviewers comment: 4) It would be beneficial to develop an electrochemical model based on the computational results to describe the energetics of the process (Possibly not in this study, but at least recommendations for further investigation could be added in the discussion part)
Authors reply: We thank the reviewer for the suggestion and we have added a sentence to the revised manuscript pointing out this interesting research direction (page 10 line 308):
“... which can be of importance for applications in catalysis. AIMD simulations can also guide development of electrochemical models.”
Reviewers comment: 5) In principle, it would be useful to include ideas for experimental verification of the results in the discussion section. The study currently focuses on a comprehensive analysis of the model, but at times it lacks connections to real experimental data, which would strengthen the validity of the findings. Could in situ XAS, for example (or other methods) be used to verify the results?
Authors reply: We agree with the reviewer that it would be very interesting to compare our simulation results to experiments. However, there are several challenges associated with experimental investigation TiO2-water interfaces. First the experimental signal coming from the interface is often weaker than that coming from the bulk sample. Furthermore, reactive events involved in water surface chemistry on TiO2 takes place on a very short time scale ~0.1 ps which is too fast for many experimental techniques. We have referenced several STM experiments but they are usually performed under vacuum conditions and our study is concerned with hydrated surfaces. A direct comparison between simulations and experiments is therefore challenging. However, as the reviewer points out, in situ spectroscopy experiments can provide important insights allowing for comparison with our work. For example, Walle et al. [Walle et al, J. Phys. Chem. C, 2011, 115, 9545-9550] used surface-sensitive photoelectron spectroscopy to study water adsorption/dissociation on anatase (101). Although their study did not provide detailed atomistic insight to water dissociation at oxygen vacancy sites it supported a picture were water molecules adsorb both molecularly and dissociatively. This is in agreement with our simulation results and we have therefore added the following statement to the discussion section (page 9, line 280-282):
“Our simulation results supports a mixed picture were water molecules both adsorb molecularly and dissociatively to the anatase (101) surface which is in agreement with surface-sensitive photoelectron spectroscopy experiments.”[Walle et al, J. Phys. Chem. C, 2011, 115, 9545-9550]
Reviewers comment: 6) line 135: “In order to neutralize the system 11 Na+ ions were added”. This is a new approach for me. Please justify its validity in the text. To what extent could this approach introduce artifacts? Why can't H⁺ (not Na+) ions be added?
Authors reply: We thank the reviewer for the question regarding ions. Since simulations are carried out with periodic boundary conditions, the total charge must be zero and with the force field used in this work adding Na+ ions is the standard procedure for neutralizing the system. In principle the same could be done by adding H+ ions but this is not a good option since it would correspond to acidic conditions and we want our simulation to represent neutral pH. It can also be noted that experiments typically use buffer solutions to control pH and this also adds counter ions to the system. We use periodic boundary conditions and calculate electrostatic interactions using the particle mesh Ewald (PME) algorithm. Given the box size used in the simulation and the chosen PME parameters we do not expect artifacts.
Reviewer 2 Report
Comments and Suggestions for Authors
In this work, the authors studied the hydrated anatase (101) titanium dioxide surfaces with oxygen vacancies using a combination of classical and ab initio molecular dynamics simulations. The results are rich and the discussion is thorough. The manuscript can be accepted after a minor revision.
1. The conclusion section can be appropriately shortened.
2. Can the author provide input files for modeling, so that other researchers can repeat the theoretical work.
Author Response
Reviewers comment: In this work, the authors studied the hydrated anatase (101) titanium dioxide surfaces with oxygen vacancies using a combination of classical and ab initio molecular dynamics simulations. The results are rich and the discussion is thorough. The manuscript can be accepted after a minor revision.
1. The conclusion section can be appropriately shortened.
Authors reply: We thank the reviewer for evaluating our work as rich and thorough. We agree that some parts of the conclusion section can be written in a briefer way. To address this we have removed the following sentences from the conclusion section:
“It was investigated if oxygen vacancies on this surface become stabilized by molecularly adsorbed water or if they undergo reaction with water leading to hydroxylation.”
we also removed:
“It was also investigated how this process affects the structure and dynamics of water near the surface.”
We have also reformulated the sentence:
“Results from classical MD simulations are highly dependent on the empirical representation of molecular interactions and in order to gain further confidence future works can investigate the effect of using different force fields.”
to the briefer version (page 10, line 330-332):
“Results from classical MD simulations are highly dependent on the force field used and in order to gain further confidence future works can investigate the effect of using different models.”
Reviewers comment: 2. Can the author provide input files for modeling, so that other researchers can repeat the theoretical work.
Authors reply: This is valuable suggestion from the reviewer, and we included in the revised manuscript archive of files necessary for running ab initio and classical MD simulations as a part of the Supporting Information.
Reviewer 3 Report
Comments and Suggestions for Authors
The research article ‘Oxygen vacancies on hydrated anatase (101) surfaces: Insights from classical and ab initio molecular dynamics simulations’
In this research, hydrated anatase (101) titanium dioxide surfaces with oxygen vacancies have been studied using a combination of classical and ab initio molecular dynamics simulations. The Authors made an interesting work, which can cover some gaps in the science of titanium oxides and suboxides; however, this gap was not touched in the manuscript. There are some remarks for the authors:
1. The calculation part of this work is interesting, but where is the connection to the real experiments? Can the calculations presented in this manuscript explain the results of some research articles?
2. The current focus of the manuscript is narrow. To broaden the audience, it should be more connected to materials science and practical applications, where the calculated effects take place.
3. The introduction part is not accurate at many points. For example:
· “This metal oxide occurs in three polymorphic forms: rutile, anatase and brookite.” There are way more polymorphic forms of titania and they are discovered 80 years ago. See: https://doi.org/10.3390/cryst14070647 and https://doi.org/10.3390/catal8120601.
· “Applications of TiO2 include solar cells[ 4], white pigment[5] in paints and cosmetics, UV-filter in sunscreen lotions[ 6 ] as well as self-cleaning surface coatings.” – There are way more application of titania and they should be listed in the introduction.
· “TiO2 has also received a significant amount of interest in the context of photocatalytic water splitting as a way of producing hydrogen from water using clean energy from sunlight. – it is not true, as all TiO2 polymorphs have a bandgap of higher than 3 eV. The photocatalytic activity of these polymorphs is shown only under UV; however, some titanium suboxides possess a narrow bandgap suitable for photocatalytic water splitting under sunlight.
· “Some titanium dioxide surfaces have been reported to have oxygen vacancies on as much as 15% of all O2c sites.” – Such a high rate of vacancies leads to the formation of titanium suboxides/black titania.
4. How does this study comply with other works of titanium suboxides Magneli phases and Black titania?
In general, Authors should find the connection between their calculations and the practical experiments they have performed and published.
Author Response
Reviewers comment: In this research, hydrated anatase (101) titanium dioxide surfaces with oxygen vacancies have been studied using a combination of classical and ab initio molecular dynamics simulations. The Authors made an interesting work, which can cover some gaps in the science of titanium oxides and suboxides; however, this gap was not touched in the manuscript. There are some remarks for the authors:
1. The calculation part of this work is interesting, but where is the connection to the real experiments? Can the calculations presented in this manuscript explain the results of some research articles?
Authors reply: We thank the reviewer for evaluating our work as interesting. We agree that making connection to experimental studies is important and this question also closely connects to question 5 by Reviewer 1.
There are several challenges associated with experimental studies of TiO2-water interfaces. For example, reactive events take place on a very short time scale ~0.1 ps which is too fast for many experimental techniques. Furthermore, signals coming from the interface region are often weak in comparison to those from the bulk region. For these reasons a direct comparison between simulations and experiments is often challenging. We have cited several STM experiments but they were carried out under vacuum conditions and our study is focused on hydrated interfaces. We think that surface-sensitive spectroscopy techniques is a promising source of information for hydrated surfaces that can be compared with our simulations. For example, Walle et al. [Walle et al, J. Phys. Chem. C, 2011, 115, 9545-9550] used surface-sensitive photoelectron spectroscopy to study water adsorption/dissociation on anatase (101). Although their study did not provide detailed atomistic insight to water dissociation at oxygen vacancy sites it supported a picture were water molecules adsorb both molecularly and dissociatively to anatase (101). This is in agreement with our results and we have therefore added the following to the discussion section (page 9, line 280-282):
“Our simulation results supports a mixed picture were water molecules both adsorb molecularly and dissociatively to the anatase (101) surface which is in agreement with surface-sensitive photoelectron spectroscopy experiments.”[Walle et al, J. Phys. Chem. C, 2011, 115, 9545-9550]
Reviewers comment: 2. The current focus of the manuscript is narrow. To broaden the audience, it should be more connected to materials science and practical applications, where the calculated effects take place.
Authors reply: The reviewer points out that the topic of our study is quite specialized. We agree with the reviewer and we have tried to broaden the audience by describing more how the work fits into a bigger picture and how it connects to specific applications as well as other scientific challenges. Specifically we have reformulated the following sentence (page 3, line 101-104):
“Obtaining a detailed picture of water structure and surface chemistry at titanium
dioxide-water interfaces in general, and the anatase (101) surface in particular, is of scientific
interest but also of importance for understanding the catalytic processes taking place in
these systems.”
to make the appeal broader we have replaced it with:
“Obtaining a detailed picture of water structure and surface chemistry at titanium dioxide-water interfaces in general, and the anatase (101) surface in particular, is a major challenge in several scientific fields including the modelling, surface characterization and synthesis communities. It is also crucial for better understanding the catalytic processes taking place in these systems, particularly photocatalytic water splitting and other applications.”
Our work can also be of interest for people studying the effect of nanomaterials on biological systems as well as on the environment. We have therefore added (page 3, line 106-107):
“It is also important for understanding the effect of these materials on biological systems as well as on environment where they exist in aqueous solution to ensure their safety.”
We have also added the sentence (page 3, line 107-109):
“Here simulations can provide important atomistic information that can complement and help interpretation of experimental studies.”
Reviewers comment: 3. The introduction part is not accurate at many points. For example:
· “This metal oxide occurs in three polymorphic forms: rutile, anatase and brookite.” There are way more polymorphic forms of titania and they are discovered 80 years ago. See: https://doi.org/10.3390/cryst14070647 and https://doi.org/10.3390/catal8120601.
Authors reply: We thank the reviewer for bringing to our attention that certain points in the introduction need to be corrected or made more clear. The reviewer is right that rutile, anatase and brookite are not the only polymorphic forms of TiO2 but they are the most common forms. To address this comment we have added the references suggested by the reviewer and reformulated the sentence (page 1, line 21-22):
“This metal oxide occurs in three polymorphic forms: rutile, anatase and brookite.”
to
“This metal oxide occurs in several polymorphic forms[Jayashree, Swaminathan and Ashokkumar, Meiyazhagan, Catalysts, 8, 2018; Ramanavicius, Simonas and Jagminas, Arunas, Crystals, 14, 2024] and the most common ones are: rutile, anatase and brookite.”
Reviewers comment:· “Applications of TiO2 include solar cells[ 4], white pigment[5] in paints and cosmetics, UV-filter in sunscreen lotions[ 6 ] as well as self-cleaning surface coatings.” – There are way more application of titania and they should be listed in the introduction.
Authors reply: The reviewer is right that TiO2 has many applications. We have mentioned several of them in the introduction of the manuscript but we cannot list all. To address the reviewer’s comment we have added references to two reviews on TiO2 applications and also added the following sentence (page 1, line 30-31):
“Applications of TiO2 has been subject to several extensive reviews.[Kazuya Nakata and Akira Fujishima, Journal of photochemistry and photobiology C: Photochemistry Reviews, 13, 3, 2012, 169-189; Francesco Parrino and Leonardo Palmisano, ‘Titanium dioxide (TiO2) and its applications’, 2020, Elsevier]”
Reviewers comment:· “TiO2 has also received a significant amount of interest in the context of photocatalytic water splitting as a way of producing hydrogen from water using clean energy from sunlight. – it is not true, as all TiO2 polymorphs have a bandgap of higher than 3 eV. The photocatalytic activity of these polymorphs is shown only under UV; however, some titanium suboxides possess a narrow bandgap suitable for photocatalytic water splitting under sunlight.
Authors reply: We thank the reviewer for this valuable comment. We agree that pure TiO2 is not suitable for photocatalytic water splitting by visual light due to the wide band gap (> 3 eV) making it adsorb in the UV range. However, the band gap can be shifted to the more intense VIS part of the solar spectrum, e.g. by doping or by introducing defects. To make this more clear we have added the following sentence to the introduction of the revised manuscript (page 1, line 28-30):
“TiO2 has also received a significant amount of interest in the context of photocatalytic water splitting as a way of producing hydrogen from water using clean energy from sunlight. However, the wide band gap of TiO2 (>3 eV) makes it adsorb in the UV range and to make water splitting more efficient it needs to be shifted to the more intense VIS part of the solar spectrum, e.g by doping or introducing defects.[Dozzi, Maria Vittoria and Selli, Elena, Journal of Photochemistry and Photobiology C: Photochemistry Reviews, 14, 2013, 13-28]”
Reviewers comment:· “Some titanium dioxide surfaces have been reported to have oxygen vacancies on as much as 15% of all O2c sites.” – Such a high rate of vacancies leads to the formation of titanium suboxides/black titania.
Authors reply: The reviewer has a good point that we should mention/discuss titanium suboxides/black TiO2. We have therefore added the following sentence to the revised manuscript (page 2, line 44-46):
“Some titanium dioxide surfaces have been reported to have oxygen vacancies on as much as 15% of all O2c sites. Oxygen deficiency leads to formation of titanium suboxides, e.g. Magneli phases, and these materials have received significant interest due to their electrochemical and photocatalytic properties.[Kumar, Ashish and Barbhuiya, Najmul H and Singh, Swatantra P, Chemosphere, 307, 2022, 135878]”
Reviewers comment: 4. How does this study comply with other works of titanium suboxides Magneli phases and Black titania?
Authors reply: This question connects to the previous comment by the reviewer about titanium suboxides and Magneli phases. We have mentioned about titanium suboxides and Magneli phases in the revised version of the manuscript and provided references for readers who wish to find out more about these materials. However, in our work we did not observe formation such phases. In fact our ab initio simulation showed that surface oxygen vacancies on hydrated surfaces react with water to hydroxylate the surface, thereby eliminating the oxygen vacancy. In our classical MD simulation oxygen vacancies were only present on the surface.
Reviewers comment: In general, Authors should find the connection between their calculations and the practical experiments they have performed and published.
Authors reply: We agree with the reviewer that our work can be improved by making connection to experiments. The question is closely connected to question 1 by the reviewer and was also pointed out by reviewer 1 (question 5). As we explained in answers to those questions there are several challenges associated with experimental investigation TiO2-water interfaces. For example, the experimental signal coming from the interface is often weaker than that coming from the bulk sample. Furthermore, reactive events involved in water surface chemistry on TiO2 takes place on a very short time scale ~0.1 ps which is too fast for many experimental techniques. To connect our work to experiments we have referenced several STM studies but as discussed in the paper they are performed under vacuum conditions and our work is concerned with hydrated surfaces. A direct comparison between simulations and experiments is therefore challenging. However, surface-sensitive spectroscopy experiments can provide important insights allowing for comparison with our work. For example, Walle et al. [Walle et al, J. Phys. Chem. C, 2011, 115, 9545-9550] used surface-sensitive photoelectron spectroscopy to study water adsorption/dissociation on anatase (101). Their study did not provide a detailed atomistic picture of water dissociation at oxygen vacancy sites but it supported that water molecules adsorb both molecularly and dissociatively on anatase (101). This is in agreement with our results and we have pointed it out in the following sentence in the discussion section of the revised manuscript (page 9, line 280-282):
“Our simulation results supports a mixed picture were water molecules both adsorb molecularly and dissociatively to the anatase (101) surface which is in agreement with surface-sensitive photoelectron spectroscopy experiments.”[Walle et al, J. Phys. Chem. C, 2011, 115, 9545-9550]
Round 2
Reviewer 1 Report
Comments and Suggestions for Authors
The revision has been done well. I recommend two corrections:
-
The discussion about the motivation for choosing Na⁺ ions for system neutralization (Q6) is valuable and not entirely trivial. It would be beneficial to incorporate this explanation in the main text or the Supplementary Information (SI).
-
The response regarding the electrochemical model (Q4) remains rather formal. Either a more in-depth analysis should be provided (e.g., specifying which intermediates need to be calculated) or the sentence should be removed, as it does not sufficiently address the reviewer’s comment in its current form.
Author Response
Reviewer's comment
The revision has been done well. I recommend two corrections:
1.The discussion about the motivation for choosing Na⁺ ions for system neutralization (Q6) is valuable and not entirely trivial. It would be beneficial to incorporate this explanation in the main text or the Supplementary Information (SI).
Authors reply: We thank the reviewer for commenting that our revision has been done well. Regarding the comment about adding Na+ ions we have included the following sentence in the main text of the revised manuscript explaining/motivating the neutralization procedure (see page 4, line 148-149):
"In order to neutralize the system 11 Na+ ions were added. These ions are present both in biological systems and in buffer solutions keeping neutral pH."
Reviewer's comment
2.The response regarding the electrochemical model (Q4) remains rather formal. Either a more in-depth analysis should be provided (e.g., specifying which intermediates need to be calculated) or the sentence should be removed, as it does not sufficiently address the reviewer’s comment in its current form.
Authors reply: We appreciate the comment regarding electrochemical models. This topic was brought to our attention by the reviewer in the first revision and we therefore added a statement pointing it out as a future research direction. However, an in-depth analysis and subsequent development of electrochemical models is out of the scope of the present study and we have therefore followed the second recommendation by the reviewer and removed the statement:
"AIMD simulations can also guide development of electrochemical models."
Reviewer 3 Report
Comments and Suggestions for Authors
Dear Authors,
thank you for the updates on the manuscript. It can be proposed for publication in the present form.
Author Response
Reviewer's comment
Dear Authors,
thank you for the updates on the manuscript. It can be proposed for publication in the present form.
Authors' reply: We thank the reviewer for proposing publication of our work in its present form.